# Evaluation of the Nutrient Composition, In Vitro Fermentation Characteristics, and In Situ Degradability of *Amaranthus caudatus*, *Amaranthus cruentus*, and *Amaranthus hypochondriacus* in Cattle

**DOI:** 10.3390/ani11010018

**Published:** 2020-12-24

**Authors:** Kim Margarette C. Nogoy, Jia Yu, Young Gyu Song, Shida Li, Jong-Wook Chung, Seong Ho Choi

**Affiliations:** 1Department of Animal Science, Chungbuk National University, Cheongju, Chungbuk 28644, Korea; nogoykimmargarette@gmail.com (K.M.C.N.); 2018139002@chungbuk.ac.kr (J.Y.); trevi92@naver.com (Y.G.S.); lishida@chungbuk.ac.kr (S.L.); 2Department of Industrial Plant Science and Technology, Chungbuk National University, Cheongju, Chungbuk 28644, Korea

**Keywords:** amaranth, caudatus, cruentus, hypochondriacus, in vitro, in situ

## Abstract

**Simple Summary:**

The amaranth plants, one of the crops that can grow in poor soil and areas with extreme weather conditions—high temperature and limited rainfall—showed high potential feed value as forage for ruminants. An extensive study will help extend its utilization as an alternative protein and fiber feed source in cattle feeding. In this study, the nutrient compositions of the three different species of amaranth, *Amaranthus caudatus*, *A. cruentus*, and *A. hypochondriacus*, were evaluated. Rumen fluid was incubated with the amaranth plants to evaluate fermentation characteristics (total gas production, total volatile fatty acids concentration, pH, and ammonia concentration). There were no differences among the different amaranth species, but all amaranth showed favorable fermentation values. The effective degradabilities of dry matter and crude protein of the amaranth forages were also determined. Compared to other studies, the effective degradabilities of dry matter (33–56%) and crude protein (27–59%) of the amaranth were lower; these results could be due to the maturity stage at which the forages were harvested. In terms of chemical composition, the amaranth forages showed better nutritive value than the locally produced forages in Chungcheong province of Korea. The amaranth forages showed 11.95–14.19% crude protein (CP), 45.53–70.88% neutral detergent fiber (NDF), and 34.17–49.83% acid detergent fiber (ADF) contents. The high nutrient composition, highly effective degradability of dry matter, and crude protein coupled with the favorable fermentation characteristics suggest that the amaranth forages showed good to excellent feed quality to cattle.

**Abstract:**

The amaranth plants showed high potential feed value as forage for ruminants. An in-depth study of this plant, particularly in cattle, will help extend its utilization as an alternative protein and fiber feed source in cattle feeding. In this study, the nutrient compositions of three different species of amaranth, *Amaranthus caudatus L.*, *Amaranthus cruentus L.*, and *Amaranthus hypochondriacus L.*—two varieties for each species, A.ca 74, A.ca 91, A.cu 62, A.cu 66, A. hy 30, and A. hy 48—were evaluated. The in vitro technique was used to evaluate the fermentation characteristics such as total gas production, total volatile fatty acids (VFA) concentration, pH, and ammonia concentration of the rumen fluid. Moreover, the effective degradabilities of dry matter (EDDM) and crude protein (EDCP) of the amaranth forages were determined through in situ bag technique. The amaranth forages: *A. caudatus*, *A. cruentus*, and *A. hypochondriacus* showed better nutritive value than the locally produced forages in Chungcheong province of Korea. The CP of the amaranth ranged from 11.95% to 14.19%, and the neutral detergent fiber (NDF) and acid detergent fiber (ADF) contents ranged from 45.53% to 70.88% and 34.17% to 49.83%, respectively. Among the amaranth varieties, *A. hypochondriacus* 48 showed the most excellent ruminant feed nutrient quality (CP, 14.19%; NDF, 45.53%; and ADF, 34.17%). The effective degradabilities of dry matter (EDDM; 33–56%) and crude protein EDCP (27–59%) of the amaranth were lower compared to other studies, which could be due to the maturity stage at which the forages were harvested. Nonetheless, *A. hypochondriacus 48* showed the highest EDDM (56.73%) and EDCP (59.09%). The different amaranth species did not differ greatly in terms of total VFA concentration or molar proportions, total gas production, or ammonia-N concentration. The high nutrient composition, and highly effective degradability of dry matter and crude protein, coupled with the favorable fermentation characteristics, suggest that the amaranth forages showed good to excellent feed quality for cattle.

## 1. Introduction

The amaranth (*Amaranthus* spp.) is one of the crops that can grow in poor soil and areas with extreme weather conditions—high temperature and limited rainfall. These characteristics make it a valuable plant product, particularly to parts of the world with shortages of water resources. Moreover, the amaranth leaves and stems contain natural antioxidants such as vitamin C, phenolic acids, and flavonoids [1,2] that could increase color stability in meat. Amaranth forages are excellent sources of fiber and protein, which makes the amaranth a good alternative feed to livestock. The amaranth has been long ago studied and used for livestock feeding and recently arose in ruminant livestock production when Peiretti [3] reported the highly beneficial nutrients and good fatty acids content the amaranth has, which potentially increase its value as a feedstuff in ruminants. The nutritive value, digestibility, and acceptability of amaranth to ruminants could be affected by ensiling or processing [4,5], while Seguin et al. [6] argued that fresh and ensiled amaranth are both highly degradable in the rumen. Aside from the processing methods, the species of the amaranth forage could also largely affect the value of the amaranth as feedstuff to ruminants. The *Amaranth hypochondriacus* grain can be used as a partial substitute for barley in sheep diets [7], and either fresh or ensiled with molasses has potential value as a ruminant feedstuff [8]. The *Amaranth cruentus* also showed good potential as an energy source for growing lambs but did not affect the weight gain or feed utilization [9]. Studies about *Amaranth caudatus* showed its value as feedstuff in rabbits and poultry nutrition.

Regarding these studies, the amaranth showed high potential value as forage for ruminants, and studying it extensively, particularly in cattle, will help increase our understanding of its utilization and extend its potentialities as feed in cattle nutrition as an alternative protein and fiber source. In this study, we aimed to determine the nutritional value of the three different species of amaranth—namely, *Amaranthus caudatus L.*, *Amaranthus cruentus L.*, and *Amaranthus hypochondriacus L.*, and the fermentation characteristics of the amaranth species through in vitro technology and an in situ, nylon bag study model. Specifically, we aimed to determine the nutrient compositions of the different amaranth species, total gas production, total volatile fatty acids (VFA) concentration, pH, and ammonia concentration of the rumen fluid incubated in vitro, and the effective degradabilities of dry matter (EDDM) and crude protein of the amaranth incubated in situ.

## 2. Materials and Methods

### 2.1. Sample Preparation and Nutrient Composition Analysis

The amaranth species studied were *A. caudatus* (A.ca), *A. cruentus* (A.cu), and *A. hypochondriacus* (A. hy) with two varieties for each species: A.ca 74, A.ca 91, A.cu 62, A.cu 66, A. hy 30, and A. hy 48. Original accession sources of A.ca, A.cu, and A. hy were Peru, Rome, and China, respectively. The six different amaranth plants were planted at the same seeding date in the experimental field in Chungbuk National University, South Korea, and grown under the recommended compost dosages, fertilizers, and relevant cultural practices. The plants were harvested after 120 days of sowing the seed and were cut wholly (leaf, stem, and seed head) from the 5 cm stubble height and chopped into fragments of 3 cm length using a manual cutter. The samples were then weighed and dried at 65 °C for 72 h through a forced-air drying oven (Machine type OF-22GW, JEIO TECH) to a constant weight to analyze the DM contents, and then ground through a Cyclotec mill (Foss Tecator Cyclotec 1093) using a 1 mm screen before analysis. The nutrient composition of the samples was analyzed as follows: crude protein (CP) calculated as N × 6.25 (AOAC Method 2001.11), ether extract (EE) (AOAC Official Method 2003.05), and ash (AOAC Official Method 942.05) based on Official Methods of Analysis by AOAC [10]; neutral detergent fiber (NDF) and acid detergent fiber (ADF) were according to the method of Goering and Van Soest [11].

### 2.2. In Vitro Incubation Procedure

To determine the fermentation characteristics, such as pH, ammonia-N concentration, and total gas production of amaranth plants, the samples were first incubated in vitro with rumen fluid. The in vitro incubation was conducted according to the procedure described by Getachew et al. [12]. Rumen contents were obtained 2 h after the morning feeding (08:00) from two ruminal-cannulated non-lactating Korean native cows (Hanwoo) fed 10 kg/d total diets daily (6 kg concentrate and 4 kg ryegrass, as fed basis), twice per day in an equal volume. The rumen fluid was hand-squeezed and filtered through 8 layers of cheesecloth and kept in a water bath at 39 °C. Incubation solution was prepared by mixing 50 mL filtered rumen fluid with 100 mL McDougall’s artificial saliva [13] in a 250 mL Erlenmeyer flask (Pyrex, Sigma-Aldrich). The artificial saliva was prepared prior to the experiment and was kept warm at 39 °C. One nylon bag contained one gram of dried and ground amaranth sample (weight per surface area: 40 mg/cm^2^) and was incubated in the flasks for 3, 6, 12, 24, and 48 h. The flasks were then sealed with silicone rubber stoppers with 3-way stopcocks and were incubated anaerobically in an orbital shaking incubator (Model VS-8480, Vision Scientific) at a speed of 135 rpm up to 24 h at 39 °C. Carbon dioxide (CO_2_) was continuously flushed throughout the operations. Incubation was stopped by removing the bottles from the shaking incubator at 3, 6, 12, 24, and 48 h. Gas was read by inserting a 50mL calibrated glass syringe (Fortuna, Sigma-Aldrich) into the 3-way stopcock, and the pH of the incubated solution was also immediately measured using a tabletop pH meter (Hanna Instruments Edge Dedicated pH/ORP meter HI2002). At the same time, an aliquot of incubated solution (two 1 mL) was collected from each flask for ammonia and volatile fatty acid (VFA) analysis. The 1 mL aliquot for ammonia analysis was mixed with 0.2 mL phosphoric acid to stop the fermentation. Ammonia concentration was determined using a spectrophotometer (Optizen 3220UV) [14]. The other 1 mL aliquot was mixed with the 0.02 mL pivalic acid solution as the internal standard for the VFA analysis as described by Li et al. [15]. The in vitro incubation was conducted 3 times with each treatment duplicated each time under similar conditions. For each run time, two flasks were designated for each of the 6 treatments (amaranth variety) per incubation period. There were 6 treatments × 2 flasks × 5 incubation period so that there were sixty (60) flasks in every run time of in vitro incubation. Each treatment had 2 replicates × 3 run times totaling to 6 observations per parameter per incubation period.

### 2.3. In Situ Incubation Procedure

To determine the effective degradability of dry matter and crude protein of the amaranth forages, another set of amaranth samples were prepared in nylon bags for in situ incubation. The in situ study was conducted in the same animals used for the in vitro study: two ruminal-cannulated non-lactating Korean native cows (Hanwoo) (268 ± 8 kg) were fed 10 kg/d total diets daily (6 kg concentrate and 4 kg ryegrass, as fed basis), twice per day in an equal volume. The animals were adapted to the diet for 7 days prior to study and water was provided ad libitum throughout the trial. Six sets (1 set = 1 amaranth type) of duplicate nylon bags containing 2 g samples of dried and ground amaranth samples were prepared. The nylon bag size was 5 × 5 cm with a pore size of 50 µm. The nylon bags were placed into net bags (about 50 cm length) according to their incubation time, tied in a rope, and were fixed inside the rumen. The bags were placed 1 h after the morning feeding. One bag containing 6 sets of duplicated samples from each Hanwoo was removed after 3, 6, 12, 24, and 48 h incubation inside the rumen. Upon removal, nylon bags were rinsed immediately under tap water with subsequent washing in a tub until the rinsing water appeared clear. The samples were dried at 80 °C for 48 h in the drying oven to constant weight to measure DM and CP degradation. The degradability at 0 incubation time was obtained by rinsing unincubated nylon bag samples. Additional triplicate bags were also incubated in autoclaved rumen fluid for 0.5 h to correct for washing losses. The in situ study was conducted 3 times with each treatment (three different amaranth species × two varieties each species) duplicated each incubation time under similar conditions. Each treatment had 2 replicates × 3 run times totaling 6 observations per parameter per in situ incubation period.

### 2.4. Estimation of Effective Degradability In Situ

The dry matter losses (parameters a, b, c) for each incubation time were calculated from the portion remaining after incubation and was fitted to the equation of [16] as follows: Y(t) = [a + b(1 − ect)] where, Y(t) = proportion of the incubated material degraded at time t; t = incubation time (h); a = highly soluble and instantly degradable fraction; b = insoluble and slowly degradable fraction; c = rate constant of degradation (%h^−1^); and e = 2.7182 (base for natural logarithm) through the Marquardt iterative procedure using the PROC NLIN of SAS (SAS Software 9.2). The fitted equation parameters a, b, and c were then used to calculate the effective degradability of DM (EDDM) and CP (EDCP) of the amaranth samples using the following equation: ED = a + (b × c)/(c + r) where r is the rate constant of passage (%h^−1^) and a hypothetical of 0.023/h passage rate. The 0.023/h passage rate of digesta through the rumen was used as it was the best-fit rate of passage irrespective of the forage types according to the meta-analysis of Krizsan et al. [17].

### 2.5. Statistical Analysis

The study was conducted as a 3 × 2 factorial design representing three species of amaranth (*A. caudatus*, A.ca; *A. cruentus*, A.cu; and *A. hypochondriacus*, A. hy) and two varieties for each amaranth specie summing to 6 amaranth samples (A.ca 74, A.ca 91, A.cu 62, A.cu 66, A. hy 30, and A. hy 48). Each of the six amaranth samples was replicated twice per incubation time, and in vitro and in situ procedures were repeated three times. For each variable measured at each time, replicates were averaged, and the total number of observations was 6 (treatments) × 3 (times) = 18 observations. The 18 observations were subjected to least square analysis of variance ANOVA by using the GLM procedure in SAS. Significances were declared at *p* < 0.05, and if differences were detected, data were further subjected to Duncan’s multiple range test (DMRT). Data are presented as means ± standard deviations instead of mean values and SEMs to indicate dispersion of the data from mean, thereby presenting the result of this study more precisely.

## 3. Results

### 3.1. Chemical Compositions of Different Amaranth Forages

The chemical compositions of the amaranth forages are shown in Table 1, and only the main effects are shown, as no significant interaction occurred in the statistical analysis. All the nutrient contents except EE and ash of the amaranth forages significantly differed among species. The moisture content (MC) and crude protein (CP) contents were significantly higher in *A. hypochondriacus* (A. hy) compared to *A. caudatus* (A.ca) and *A. cruentus* (A.cu). The NDF was significantly higher in *A. cruentus* than in *A. caudatus* and *A. hypochondriacus*, and the ADF was significantly higher in *A. caudatus* compared to the other two amaranth species. Varieties of the amaranth plants showed significant differences in all nutrient contents except crude fat (EE). The MC was significantly the highest in A. hy 48 than other amaranth plants (*p* < 0.0001). The CP content was significantly the highest in A. hy 48 and A. hy 30 followed by A.ca 74 and A.cu 64, leaving A.cu 66 and A.ca 91 having the lowest CP contents (*p* < 0.05). The NDF content was significantly the highest in A.cu 66. The varieties A.cu 62 and A. hy 30 also showed significantly high NDF content but were comparable to A.ca 91 and A.ca 74; A. hy 48 showed the lowest NDF content among all varieties of amaranth forages (*p* < 0.01). The ADF content was significantly the highest in A.cu 66 and A. hy 30 followed by A.cu 62 and A.ca 91 and then by A.ca 74 and A. hy 48 (*p* < 0.05). Ash content was found highest to lowest in this ranking A. hy 48 > A.cu 66> A.ca 91 > A.ca 74 > A.cu 62 > A. hy 30 (*p* < 0.0001).

### 3.2. In Vitro Fermentation Characteristics of Different Amaranth Forages

The effects of different amaranth forages on the pH, ammonia-N concentration, and total gas production when incubated in vitro in rumen fluid are shown in Table 2. The main effects only are shown in the table, as no significant interaction effects occurred. The pH of the rumen fluid incubated with amaranth forages decreased with increasing incubation time. The ammonia-N concentration increased with increasing incubation time, except that of the *A. cruentus*. Both varieties of *A. cruentus*, A.cu 62 and A.cu 66, showed the highest ammonia-N concentration at 6 h but had become significantly the lowest at 24 and 48 h of incubation time. The varieties *A. hypochondriacus* (A. hy 30 and A. hy 48) showed the second highest ammonia-N concentration at 6 and 24 h incubation and showed the highest ammonia-N concentration at 48 h incubation time. The ammonia-N concentration of *A. caudatus* was lowest at 6 h, highest at 24 h, and second highest at 48 h of incubation time. In terms of pH values, *A. cruentus* showed the highest pH, and *A. caudatus* showed the lowest at all incubation times. The pH values of A.cu 66 and A.cu 62 ranged from 5.96 to 6.28; those of A. hy 30 and A. hy 48 ranged from 5.74 to 6.23; and for A.ca 91 and A.ca 74 pH ranged from 5.63 to 6.14. The total gas production of the rumen fluid incubated with amaranth forages also increased with increasing incubation time (Table 2). Remarkably, *A. cruentus* showed the highest total gas production as compared to *A. caudatus* and *A. hypochondriacus*.

The total VFA concentration and VFA molar proportions were shown in Table 3. Total VFA concentration at all incubation times except at 12 h was unaffected by the different amaranth forages. The total VFA concentration of *A. caudatus* was significantly the highest while *A. cruentus* was significantly the lowest among the amaranth species (*p* < 0.05). A significant effect was observed in the acetate (C_2_) molar proportion at 24 h and 48 h (*p* < 0.05). The *A. hypochondriacus* showed significantly higher C_2_ molar proportion than *A. caudatus* and *A. cruentus* (*p* < 0.05). The propionate (C_3_) molar proportion was unaffected at all incubation times except at 3 h where *A. caudatus* showed increased C_3_ than *A. cruentus* and *A. hypochondriacus*. The butyrate (C_4_) molar proportion was unaffected at all incubation times except at 12 h. The *A. cruentus* showed the highest C_4_ molar proportion while *A. caudatus* showed the lowest (*p* < 0.05). The valerate (C_5_) molar proportion was significantly higher in *A. hypochondriacus* than the other two amaranth species (*p* < 0.001). The caproate (C_6_) molar proportion was significantly affected by the different amaranth species at all incubation times except at 24 h where *A. cruentus* showed significantly the highest C_6_ value than the other two amaranth species. The C2/C3 ratio was unaffected at all incubation times except at 3 h where *A. hypochondriacus* showed the highest ratio while *A. caudatus* showed the lowest (*p* < 0.05).

### 3.3. In Situ DM and CP Degradability

The main effects of the parameters a, b, c of in situ degradability and effective degradability of dry matter (EDDM) and crude protein (EDCP) were shown in Table 4. Parameters a, c, and EDDM were highest in A. hy 48. Second and third highest EDDM were observed in A.cu 62 and A.cu 66, respectively, with lower parameter a and higher parameter b than A. hy 48. The fourth highest EDDM was observed in A. hy 30 followed by A.ca 91 and A.ca 74. Parameter a of EDCP was not affected by species and subspecies while parameter b and EDCP showed significant effects. The amaranth forages A.ca 74 and A.ca 91 showed the highest parameter b (*p* < 0.001) but lowest EDCP (*p* < 0.0001). On the other hand, amaranth subspecies A. hy 30 and A. hy 48 showed the lowest parameter b (*p* < 0.001) but highest EDCP (*p* < 0.0001).

## 4. Discussion

The nutrient compositions of the different species of the amaranth forage were similar to the nutrient compositions of the locally produced forages produced in the Chungcheong province of South Korea, as reported in the study of Ki et al. [18]. The three species of amaranth, *A. caudatus*, *A. cruentus*, and *A. hypochondriacus*, showed higher crude protein than the reported protein of commonly used forages for ruminants, such as corn silage (8.4%), sudangrass (5.1%), rice straw (4.9%), and Italian ryegrass (13.4%) [18]. The three amaranths also showed lower NDF and ADF contents as compared to the locally produced forages stated. In this study, the NDF contents of the three amaranths ranged from 45.53% to 70.88%, whereas locally produced forages ranged from 60.1% to 75.6% [18]. In terms of the ADF content, the three amaranths showed a range of 34.17% to 49.83%, whereas locally produced forages showed a range of 27.3% to 64.4% [18]. The lower NDF and ADF contents of the amaranths in this study signify more non-fiber carbohydrates (NFC), such as CP, EE, ash, sugars, and starch. It was reported that the NFC in a forage closely amounts to the energy supplied to ruminants and is associated with the synthesis of the microbial protein in the rumen [19], signifying that the amaranth plants in this study contained good amounts of non-fiber nutrients for energy supply to cattle. However, in comparison to the NDF and ADF contents of mature alfalfa (52%, 42%), cornstalks (68%, 43%), and headed sorghum-sudangrass (65%, 40%) of the U.S.-Canadian Tables of Feed Composition, the amaranths in this study showed slightly higher NDF and ADF contents [20]. Among the amaranths, varieties *A. hypochondriacus 48* and *A. caudatus 74* containing 14.19–12.93% CP, 45.53–57.40% NDF, and 34.17–39.85 % ADF contents showed the most comparable feed quality with alfalfa. The variety *A. cruentus 62* which showed highs of 70.88% NDF and 49.83% ADF contents, presented the least nutritionally advantaged amaranth among the amaranth forages. Nonetheless, the *A. cruentus* showed a slightly better feed quality (12.03% CP; 70.88% NDF; 49.83% ADF) than wheat straw (CP, 4%; NDF, 85%; ADF, 54%), and is slightly comparable with the cornstalks (CP, 6%; NDF, 68%; ADF, 43%), as reported in U.S.-Canadian Tables of Feed Composition [20]. The nutrient compositions of the different amaranth species are indicative of the fact that the amaranths are nutritionally comparable to the commonly used forages for ruminants in Korea and the US/Canada.

The value of a forage has a lot to do with the energy it supplies to the ruminants through the digestibility of the fiber and non-fiber contents. The known useful methods to assess the potential value of feedstuff as energy sources for ruminants are the in situ rumen disappearance and in vitro gas production techniques. In this study, we used the rumen bag technique (in situ) to quantitatively estimate the degradability of the amaranth forages. Among the amaranth forages, the variety *A. hypochondriacus 48* showed greater parameters a and c, and lower parameter b, than the other amaranth forages, indicating that it is highly degradable in the rumen cattle. Consequently, *A. hypochondriacus 48* showed a greatly higher effective degradability of dry matter (EDDM). In the same way, although parameter a did not show a significant difference, parameters b, c, and EDCP were significantly higher in *A. hypochondriacus 48* compared to other amaranth samples. The CP content of *A. hypochondriacus 48* was high at 14.29%, signifying that higher CP content in forages is valuable in the effective degradability of CP in the rumen, as also reported by Satter [21]. The highly effective degradability of the dry matter and crude protein of *A. hypochondriacus 48* is indicative of well-balanced CP, NDF, ADF, and DM contents in the specified amaranth variety. However, although *A. hypochondriacus 48* showed the highest EDDM (56.73) in this study, it is lower than the EDDM of *A. hypochondriacus* (66.64–75.13) reported by Fazaeli and Ehsani [22]. Following the high EDDM and EDCP of *A. hypochondriacus 48* was *A. cruentus 62*, which showed 45.16 EDDM and 43.15 EDCP. The high EDCP of *A. cruentus 62* could be associated with its high CP content. The EDDM of *A. cruentus* was slightly comparable to 52.6 [23] and 55.4 [22] DM digestibility of *A. cruentus* when fed to sheep, but is lower than the 71–73 in vitro dry matter digestibility of *A. cruentus* reported by Sleugh et al. [24]. The difference in the percent effective degradability is evidently and mainly due to the in vitro technique used by Sleugh et al. (2001) [24] and the in situ technique used in this study. The *A. caudatus* showed the least EDDM and EDCP among the other amaranths. The EDDM of *A. caudatus* ranged from 33.31–34.87% which was lower than the lowest in vitro dry matter digestibility (78.10%) reported by Peiretti et al. [25]. Other than the difference regarding the method for estimating degradability, the *A. caudatus* in the study of Peiretti et al. [25] was harvested after 90 days, whereas the amaranth forages in this study were harvested after 120 days. Accumulation of cell wall contents as the plant advances in maturity dramatically decreases the degradability of plants [26,27]. Amaranth forages in this study were fully mature at 120 days when harvested, whereas amaranth forages in the cited studies were harvested at 75, 90, and 112 days (Fazaeli et al., 2011; Peiretti et al., 2018; Sleugh et al., 2001, respectively) [22,24,25]. Increased cell wall contents in the fully matured amaranth forages must have lowered the EDDM of the amaranth in this study. In comparison to the neutral detergent fiber digestibility (NDFD) of the locally produced forages in Chungcheong province of Korea [18], the EDDM of *A. cruentus* and *A. hypochondriacus* are nearly comparable to the NDFD of corn silage (56.0%), Italian ryegrass (57.7%), and Sudangrass (40.8%), and are higher than the NDFD of rice straw (23.6%).

The digestibility and energy feed quality of the amaranth forages to ruminants through evaluation of the pH, total gas production, ammonia-N, and VFA concentration were determined using the in vitro gas production technique. The DM of high-quality forages during in vitro incubation produces gas ranged from 0.37 to 0.39 mL/mg [28]. The *A. cruentus* showed higher total gas production than *A. caudatus* and *A. hypochondriacus* at all incubation times. The total gas produced by *A. cruentus* increased from 0.018 mL/mg at 3 h to 0.38 mL/mg at 48 h, whereas the species *A. hypochondriacus* produced a total gas of 0.30 mL/mg at 48 h, which is slightly lower than the range reported by Pell et al. [28]. The high total gas production could be associated with the high EDDM observed in *A. caudatus* and *A. hypochondriacus*. The increased gas production in *A. caudatus* and *A. hypochondriacus* signify that there is a sufficient accessible amount of substrate in the two amaranth forages for microbial fermentation. To support microbial synthesis for fiber digestion, the ammonia-N concentration should range from 5 to 8 mg × 100 mL^−1^ [29]. The ammonia-N concentrations of the amaranth forages at all incubation times were all within the range, except the *A. caudatus* at 3 h and 6 h, which could have contributed to its low EDDM, although ammonia-N concentration increased at 12 h through to 48 h and fell within the range. The *A. cruentus* showed the highest ammonia-N concentration at 3 h to 12 h, while *A. hypochondriacus* showed the highest at 48 h. This finding could have contributed to the high EDDM and EDCP of *A. hypochondriacus*. The pH along with the ammonia-N concentration of the rumen fluid roughly shows the fermentation activities of the forage. The pH values of 6 to 9 were reported to be the optimal pH to support the growth of cellulolytic bacteria for digestion activity [30]. Among the different species of amaranth forages, *A. cruentus* showed the highest pH values ranging from 5.96 to 6.28 at all incubation times. Following the *A. cruentus* was the *A. hypochondriacus* with a pH ranging from 5.74 to 6.23. In general, *A. cruentus* and *A. hypochondriacus* showed higher feed quality than *A. caudatus* in terms of fermentation activities of the forage incubated in vitro. *A. cruentus* and *A. hypochondriacus* showed increased pH values, total gas production, and ammonia-N concentration. Finally, the volatile fatty acid (VFA) concentration which reflects the major energy content of the feed or forage as the energy source for ruminants was not affected by different amaranth forages at any incubation time except at 12 h. The *A. caudatus* showed a higher total VFA concentration than *A. cruentus* and *A. hypochondriacus* at 12 h; however, this increase in total VFA concentration was not supported by other fermentation parameters, such as pH, gas production, ammonia-N concentration, and EDDM, which is positively related to VFA concentration [31]. Regardless of the species of the amaranth forages, the total VFA concentrations of the amaranth forages were all closely comparable with the total VFA concentration of the locally produced forages in Chungcheong province of Korea. Total VFA of the amaranth forages ranged from 59.12 to 64.94 at 48 h and was found comparable to VFA of corn silage (60.9), Italian ryegrass (55.4), and Sudangrass (50.5), and was higher than rice straw (39.3), as reported in the study of [18]. High acetate was observed in *A. hypochondriacus* at 24 and 48 h, indicating that the specified amaranth variety contained large amounts of degradable fiber in the forage. On the other hand, a significantly high propionate level was observed in *A. caudatus* at 3 h, but the amaranth forages showed comparable levels of propionate from 6 h through to 48 h. The amaranth forages did not greatly differ in terms of total VFA production and molar proportions, signifying that *A. caudatus, A. cruentus,* and *A. hypochondriacus* produced the same energy source for ruminants.

The fermentation characteristics through in vitro study and the effective degradability through in situ studies of the amaranths indicated that *Amaranthus caudatus*, *Amaranthus cruentus*, and *Amaranthus hypochondriacus* are excellent alternative feeds to cattle. The good to excellent feed quality of the amaranth on cattle could help in the utilization of uncommonly used forages.

## 5. Conclusions

In conclusion, the nutritive values, such as CP, NDF, and ADF contents of the different amaranth varieties *A. caudatus*, *A. cruentus*, and *A. hypochondriacus* were found to be closely comparable to the nutritive value of corn silage and other commonly used forages in Korea. Amaranth forages also showed better effective degradability of dry matter and crude protein than that of corn silage. The effective degradability of these nutrient contents coupled with good fermentation characteristics, such as high VFA concentration, total gas production, and ammonia-N concentration, suggest that amaranth possesses a good to excellent feed quality for ruminants, particularly for cattle. However, further research is recommended to evaluate the feed value of *Amaranth* using cattle fed with *Amaranth* forage-based diets.

## Figures and Tables

**Table 1 animals-11-00018-t001:** Chemical compositions of different amaranth forages.

Chemical Composition	*A. caudatus*		*A. cruentus*		*A. hypochondriacus*		SEM	Effects	
	A.ca 74	A.ca 91	A.cu 62	A.cu 66	A. hy 30	A. hy 48		Species	Variety
MC, %	6.12 ^b^	6.17 ^b^	6.85 ^b^	7.18 ^b^	7.03 ^b^	12.58 ^a^	0.41	***	***
EE, %	0.91	0.64	1.10	0.46	0.71	1.06	0.23	NS	NS
CP, %	12.36 ^b^	11.41 ^c^	11.93 ^b c^	11.55 ^c^	12.78 ^a^	14.29 ^a^	0.17	***	*
NDF, %	57.40 ^b^	61.67 ^b^	63.15 ^a b^	70.88 ^a^	63.90 ^a b^	45.53 ^c^	2.57	*	**
ADF, %	39.85 ^b c^	44.39 ^a b^	45.44 ^a b^	49.83 ^a^	47.53 ^a^	34.17 ^c^	2.26	*	*
Ash, %	10.25 ^b c^	11.15 ^b^	8.89 ^c^	11.17 ^b^	8.70 ^c^	13.89 ^a^	0.56	NS	***

Means in the same row with different superscripts ^a, b, c^ are significantly different. * *p* < 0.05, ** *p* < 0.01, *** *p* < 0.001. SEM, standard error of the mean; NS, not significant; MC, moisture content; EE, ether extract (crude fat); CP, crude protein; NDF, neutral detergent fiber; ADF, acid detergent fiber. Moisture contents presented in the table are dry-based.

**Table 2 animals-11-00018-t002:** The pH, ammonia-N concentration (mg/100 mL), and total gas production (mL) of amaranth forages incubated in vitro in rumen fluid.

	*A. caudatus*		*A. cruentus*		*A. hypochondriacus*		SEM	Effects	
	A.ca 74	A.ca 91	A.cu 62	A.cu 66	A. hy 30	A. hy 48			
Incubation Time (h)	pH							Species	Varieties
3	6.04 ^b^	6.09 ^b^	6.19 ^a^	6.23 ^a^	6.20 ^a^	6.17 ^a^	0.05	***	*
6	6.03 ^c^	6.14 ^b c^	6.28 ^a^	6.27 ^a^	6.23 ^a^	6.22 ^a b^	0.07	***	*
12	5.88 ^d^	5.99 ^c^	6.19 ^a b^	6.25 ^a^	6.14 ^b^	6.12 ^b^	0.07	***	*
24	5.71 ^c^	5.79 ^b c^	6.10 ^a^	6.16 ^a^	5.90 ^b^	5.87 ^b^	0.10	***	*
48	5.63 ^d^	5.69 ^c d^	5.96 ^b^	6.11 ^a^	5.74 ^c^	5.76 ^c^	0.16	***	***
	NH3-N								
3	4.47 ^b^	4.74 ^b^	7.21 ^a^	5.89 ^a b^	6.50 ^a b^	5.95 ^a b^	3.55	**	*
6	4.85 ^b^	5.28 ^b^	10.84 ^a^	12.54 ^a^	5.78 ^b^	6.61 ^b^	3.13	***	NS
12	7.61 ^a b c^	6.43 ^c^	7.16 ^b c^	9.04 ^a^	8.66 ^a b^	8.86 ^a^	2.56	***	*
24	10.24	10.56	10.61	11.25	10.10	10.65	4.26	NS	NS
48	9.92 ^a b^	7.32 ^b^	8.20 ^b^	8.27 ^b^	9.09 ^a b^	11.54 ^a^	4.06	*	*
	Gas								
3	14.83 ^b^	14.17 ^b^	31.17 ^a^	18.33 ^a^	7.50 ^b^	11.83 ^b^	8.64	***	NS
6	50.17 ^b^	49.33 ^b^	84.50 ^a^	57.33 ^a^	33.50 ^b^	45.83 ^b^	18.41	***	NS
12	110.67 ^b^	108.83 ^b^	159.50 ^a^	120.33 ^a^	90.75 ^b^	107.67 ^b^	28.84	**	NS
24	194.17 ^b^	192.00 ^b^	256.00 ^a^	214.83 ^a^	181.00 ^b^	195.17 ^b^	42.04	**	NS
48	295.33 ^b^	292.33 ^b^	381.17 ^a^	342.00 ^a^	302.00 ^b^	303.83 ^b^	59.44	**	NS

Means in the same row with different superscripts ^a, b, c, d^ are significantly different at * *p* < 0.05, ** *p* < 0.01, *** *p* < 0.0001; NS, not significant.

**Table 3 animals-11-00018-t003:** Total volatile fatty acids (VFA) concentration and VFA molar proportions of different amaranth forages.

		*A. caudatus*		*A. cruentus*		*A. hypochondriacus*		SEM	Effects	
		A.ca 74	A.ca 91	A.cu 62	A.cu 66	A. hy 30	A. hy 48		Species	Varieties
3 h										
Total VFA (mmoles/100 mL)		48.99	45.89	45.43	47.87	43.02	46.52	5.11	NS	NS
Molar proportion (mmoles/100 mmoles)	C2	59.09	58.46	58.75	59.82	59.82	59.04	1.3	NS	NS
	C3	25.66 ^a^	25.53 ^a^	24.59 ^a b^	24.42 ^b c^	23.62 ^c^	25.49 ^a b^	0.89	**	*
	C4	5.13	5.52	5.5	5.24	5.95	5.47	0.65	NS	NS
	C5	0.34	0.5	0.27	0.09	0.62	0.32	0.55	NS	NS
	C6	0.97 ^c^	1.03 ^b c^	1.21 ^a^	1.16 ^a^	1.12 ^a b^	1.02 ^b c^	0.09	***	*
	C2/C3	2.31 ^b^	2.29 ^b^	2.39 ^a b^	2.45 ^a b^	2.53 ^a^	2.32 ^b^	0.13	*	*
6 h										
Total VFA (mmoles/100 mL)		53.34	49.92	53.4	49.19	50.21	50.95	4.46	NS	NS
Molar proportion (mmoles/100 mmoles)	C2	60.05	58.67	60.15	60.29	51.15	59.55	10.31	NS	NS
	C3	25.15	25.09	24.19	24.27	23.59	25.25	0.97	NS	NS
	C4	4.71	5.08	4.71	5.1	5.02	4.92	0.45	NS	NS
	C5	0.32	0.98	0.43	0.08	0.15	0.47	0.67	NS	NS
	C6	0.96 ^c^	1.12 ^a b^	1.17 ^a^	1.10 ^a b^	1.10 ^a b^	1.04 ^b c^	0.07	*	**
	C2/C3	2.39	2.34	2.5	2.49	2.16	2.36	0.46	NS	NS
12 h										
Total VFA (mmoles/100 mL)		60.11 ^a^	59.41 ^a^	53.38 ^b^	53.22 ^b^	58.09 ^a b^	57.45 ^a b^	5.7	*	NS
Molar proportion (mmoles/100 mmoles)	C2	61.09	61.24	60.63	60.87	62.82	61.04	1.57	NS	NS
	C3	24.66	24.4	24.25	23.9	23.27	24.83	1.08	NS	NS
	C4	4.19 ^b^	4.23 ^b^	4.72 ^a^	4.73 ^a^	4.37 ^a b^	4.37 ^a b^	0.43	*	NS
	C5	0.43	0.52	0.22	0.5	0.23	0.43	0.48	NS	NS
	C6	0.93 ^b^	0.95 ^b^	1.07 ^a^	1.07 ^a^	1.00 ^b^	0.94 ^b^	0.08	**	NS
	C2/C3	2.49	2.51	2.51	2.55	2.7	2.46	0.17	NS	NS
24 h										
Total VFA (mmoles/100 mL)		59.04	58.52	62.49	57.68	60.29	66.05	5.85	NS	NS
Molar proportion (mmoles/100 mmoles)	C2	61.58 ^a b^	62.01 ^a b^	61.24 ^b^	60.81 ^b^	62.86 ^a^	61.97 ^a^	1.27	*	NS
	C3	24.16	24.16	24.64	24.76	23.39	24.64	0.96	NS	NS
	C4	4.24	4.3	4.08	4.36	4.17	3.79	0.36	NS	NS
	C5	0.56 ^b^	0.24 ^b^	0.24 ^b^	0.20 ^b^	0.68 ^a^	0.83 ^a^	0.4	**	NS
	C6	0.89	0.87	0.97	0.98	0.9	0.9	0.1	NS	NS
	C2/C3	2.56	2.57	2.49	2.46	2.69	2.52	0.14	NS	NS
48 h										
Total VFA (mmoles/100 mL)		59.12	62.94	61.67	59.42	64.94	64.33	5.13	NS	NS
Molar proportion (mmoles/100 mmoles)	C2	60.17 ^b^	60.72 ^b^	59.94 ^b^	60.55 ^b^	62.44 ^a^	60.88 ^a^	1.22	*	NS
	C3	25.13	24.94	25.92	25.12	24.41	25.89	0.9	NS	NS
	C4	4.26	4.01	4.09	4.23	3.86	3.89	0.34	NS	NS
	C5	0.33	0.66	0.41	0.53	0.34	0.52	0.44	NS	NS
	C6	0.83 ^b^	0.92 ^b^	0.95 ^a^	0.98 ^a^	0.86 ^c^	0.80 ^c^	0.07	**	NS
	C2/C3	2.4	2.44	2.32	2.41	2.56	2.35	0.11	NS	NS

Means in the same row with different superscripts ^a, b, c^ are significantly different. * *p* < 0.05, ** *p* < 0.01, *** *p* < 0.001. SEM, standard error of means; NS, not significant; C2, Acetate; C3, Propionate; C4, Butyrate; C5, Valerate; C2/C3, ratio of acetate to propionate.

**Table 4 animals-11-00018-t004:** Effective degradability of dry matter and crude protein and degradation parameters (a, b, c) of amaranth forages incubated in situ.

ED Parameters	*A. caudatus*		*A. cruentus*		*A. hypochondriacus*		SEM	Effects	
	A.ca 74	A.ca 91	A.cu 62	A.cu 66	A. hy 30	A. hy 48		Species	Subspecies
a	23.26 ^c^	14.02 ^d^	29.76 ^b^	27.48 ^b^	21.38 ^c^	37.66 ^a^	1.25	***	***
b	23.75 ^d^	40.170 ^a b^	47.59 ^a^	42.52 ^a b^	34.25 ^b c^	29.93 ^c d^	3.14	***	**
c	0.04 ^b c d^	0.05 ^b c^	0.03 ^c d^	0.02 ^d^	0.06 ^b^	0.09 ^a^	0.01	***	*
EDDM	33.31 ^f^	34.87 ^e^	45.16 ^b^	41.07 ^c^	38.86 ^d^	56.73 ^a^	0.53	***	***
a	11.34	18.28	12.77	15.94	10.96	17.55	2.37	NS	NS
b	105.65 ^a^	108.15 ^a^	69.81 ^b^	53.25 ^d^	58.96 ^c^	62.59 ^c^	1.48	**	*
c	0.01 ^d^	0.01 ^c d^	0.04 ^b c^	0.02 ^c d^	0.08 ^a b^	0.10 ^a^	0.01	***	NS
EDCP	27.90 ^e^	37.67 ^c^	43.15 ^b^	33.38 ^d^	46.86 ^b^	59.09 ^a^	1.18	***	***

EDDM, effective degradability of dry matter; EDCP, effective degradability of crude protein; a, b, c in the first column correspond to: a, highly soluble fraction of sample; b, insoluble and slowly soluble fraction of sample at time infinity; c, rate constants of degradation of fraction b.” Means in the same row with different superscripts ^a, b, c, d, e, f^ are significantly different at * *p* < 0.05, ** *p* < 0.01, *** *p* < 0.0001; NS, not significant. The rate constant of passage (%h^−1^) and a hypothetical 0.023/h passage rate were used.

## Data Availability

The data can be available from the corresponding author upon reasonable request.

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
