# Peer review of "Evaluation of the Nutrient Composition, In Vitro Fermentation Characteristics, and In Situ Degradability of Amaranthus caudatus, Amaranthus cruentus, and Amaranthus hypochondriacus in Cattle"

_animals, 2020, doi:10.3390/ani11010018_

Round 1
Reviewer 1 Report
Author have addressed properly most of the reviewers comments from previous round.
A few issues remains which a detail below.
1- The authority is missing in the scientific name. It should be reported as: Amaranthus caudatus L., A. cruentus L., and A. hypochondriacus L.
I suggest to include it at least at first appeance in abstract and introduction, however I leave the final decision with the editor.
2. The formula in lines 157 and 158 it is incorrect. Delete "-A" at the end.
Author did not corrected the formula properly.
"a" it is not equal to highly soluble and instantly degradable fraction
"b" it is not insoluble and slowly degradable fraction
"a" and "b" are only fitted equation parameters.
A= highly soluble and instantly degradable fraction
(a+b)-A = insoluble and slowly degradable fraction
Please correct accordingly
3- Author state "We've re-checked our methods and the data collected and we stand true to the result we presented. The low moisture content of the amaranth samples in our study could be due to the drying of the sample prior to chemical analysis."
Therefore, they have to clealy state in table 1 that reported moisture contents correspond to "pre-dried" samples.
In consequence, as there was not true control of pre-drying process, author must avoid any inference on Moisture content of sample.
Please correct in the manuscritp at sections were moisture is described/discussed
4 - The rate constant of passage should be specify in table 4 (possibly as foot note) to make the table self-explanatory
regards
Author Response
Kindly see below for the authors' responses to the reviewer or see attached file. Thank you very much.
Reviewer No. 1 - Comments and Suggestions for Authors
Author have addressed properly most of the reviewers comments from previous round. A few issues remains which a detail below.
General response:
Thank you for the thorough review of our manuscript. The authors greatly appreciate the reviewer’s remarks to make the manuscript acceptable for publication. The comments of the reviewer have been taken into consideration and the authors are hoping that all the changes made are of satisfaction to the reviewer. The followings are the authors point-by-point responses:
Comment 1- The authority is missing in the scientific name. It should be reported as: Amaranthus caudatus L., A. cruentus L., and A. hypochondriacus L.
I suggest to include it at least at first appeance in abstract and introduction, however I leave the final decision with the editor.
Response:
Thank you for the comment. The authors have considered the reviewer’s suggestion and the authority in the scientific names of the amaranth plants were written. Kindly refer to Line 37 of the abstract section and Line 80 of the introduction section.
Comment 2. The formula in lines 157 and 158 it is incorrect. Delete "-A" at the end.
Author did not corrected the formula properly.
"a" it is not equal to highly soluble and instantly degradable fraction
"b" it is not insoluble and slowly degradable fraction
"a" and "b" are only fitted equation parameters.
A= highly soluble and instantly degradable fraction
(a+b)-A = insoluble and slowly degradable fraction
Please correct accordingly
Response:
Thank you for the reviewer’s sound comment. As much as the authors would like to consider the reviewer’s comment, the authors stand true to the equation they have stated. Please see below as explanation:
The exponential equation of Orskov and McDonald (1979) as follows:
Y(t) = [a + b(1 – ect)]
where, Y(t) = proportion of the incubated material degraded at time t; t = incubation time (h); a = highly soluble and instantly degradable fraction; b = insoluble and slowly degradable fraction; c = rate constant of degradation (%h-1); and e = 2.7182 (base for natural logarithm)
was used to determine the dry matter losses (the parameters a, b, and c) for each incubation time through the Marquardt iterative procedure using the PROC NLIN of SAS. Then, the fitted equation parameters a, b, c were used to calculate the effective degradability of DM and CP by using the equation:
ED = a + (b × c) / (c + r)
where r is the rate constant of passage (%h-1) and a hypothetical of 0.023/h passage rate of the digesta.
For more clarity, the authors have re-written the estimation of effective degradability in situ method (Lines 158-168. In exact, the rewritten method was as below:
The dry matter losses (parameters a, b, c) for each incubation time were calculated from the portion remaining after incubation and was fitted to the equation of [16] as follows: Y(t) = [a + b(1 – ect)] where, Y(t) = proportion of the incubated material degraded at time t; t = incubation time (h); a = highly soluble and instantly degradable fraction; b = insoluble and slowly degradable fraction; c = rate constant of degradation (%h-1); and e = 2.7182 (base for natural logarithm) through the Marquardt iterative procedure using the PROC NLIN of SAS. The fitted equation parameters a, b, and c were then used to calculate the effective degradability of DM (EDDM) and CP (EDCP) of the amaranth samples using the following equation: ED = a + (b × c) / (c + r) where r is the rate constant of passage (%h-1) and a hypothetical of 0.023/h passage rate. The 0.023/h passage rate of digesta through the rumen was used as it was the best-fit rate of passage irrespective of the forage types according to the meta-analysis of Krizsan et al. [17].
The authors are hoping that the revisions made in the methods section and the explanation for the equations used in this study have cleared the confusion and were satisfactory to the reviewer.
Comment 3- Author state "We've re-checked our methods and the data collected and we stand true to the result we presented. The low moisture content of the amaranth samples in our study could be due to the drying of the sample prior to chemical analysis."
Therefore, they have to clealy state in table 1 that reported moisture contents correspond to "pre-dried" samples.
In consequence, as there was not true control of pre-drying process, author must avoid any inference on Moisture content of sample.
Please correct in the manuscritp at sections were moisture is described/discussed
Response:
Thank you for the reviewer’s comment. A brief definition of the moisture content was written at the end of the footnote of Table 1 as follows: Moisture contents presented in the table are dry-based (Line 215). Furthermore, no inferences about the moisture content of the amaranth plants were made. Results and discussion mainly pointed out the DM, CP, NDF, and ADF contents of the amaranth plants.
Comment 4 - The rate constant of passage should be specify in table 4 (possibly as foot note) to make the table self-explanatory
Response:
The authors have considered the reviewer’s comment. The rate constant of passage was specified at the end of the footnote of Table 4. Kindly refer to Line 278: Rate constant of passage (%h-1) and a hypothetical of 0.023/h passage rate were used. Thank you very much.
Reviewer 2 Report
General comments:
The study examined the species and varietal differences of six Amaranth varieties for the nutritive value, in vitro fermentation characteristics and in situ degradability of CP and DM using non-lactating Korean native cows. The experimental design was appropriate and methodologies have explained sufficiently. However, the discussion section focused more on the feed value, which was not estimated in the current experiment. Thus, this should be clarified and text relating feed value of Amaranth should be removed from the manuscript.
Specific comments:
L20: reword cattle nutrition as cattle feeding
L 34: reword the deep study as an in-depth study
L40: remove extra space between “of” and “the”
L89: remove the phrase “that were”
L102: give the appropriate reference for NDF and ADF methods used
L118: remove the term “all”
L152: remove the term “to”
L210-220 and in tables: Scientific names of Amaranth species should be in italic letters
Table 3: It would be better to mention VFA names (such as Acetate, propionate) instead of abbreviations used (C2, C3 etc)
L277-278- give appropriate reference/s
L285, 287-288- The feed value of forage is a function of nutritive value and intake. Since the estimation of the feed value of the studied amaranth species was not an objective of the current experiment, the discussion on the “feed value” of Amaranth should be excluded. Please rephrase the sentence as appropriate.
L346: Referencing style should be uniform within the text
Line363: Spelling mistake, please correct as “species”
Line370-373: Remove the discussion on antioxidant content and beef quality as those were not measured in the current study
L381: remove the conclusions on feed value
L448-453: please follow the correct referencing style
Author Response
Kindly see below point-by-point responses of the authors to the reviewer or refer to the attached file. Thank you very much.
Reviewer No. 2 - Comments and Suggestions for Authors
General comments:
The study examined the species and varietal differences of six Amaranth varieties for the nutritive value, in vitro fermentation characteristics and in situ degradability of CP and DM using non-lactating Korean native cows. The experimental design was appropriate and methodologies have explained sufficiently. However, the discussion section focused more on the feed value, which was not estimated in the current experiment. Thus, this should be clarified and text relating feed value of Amaranth should be removed from the manuscript.
General response:
The authors appreciate the sound comment of the reviewer, and hence, the authors have re-written the term “feed value” to “feed quality”, which better aligned to the objective of the study, the methods conducted, and the findings collected in this study. Kindly refer to the following lines: 33, 47, 55, 299, 302, 364, 389, and 398 where feed quality was used in replacement to feed value.
Specific comments of the reviewer have been taken into consideration and the authors are hoping that all the changes made are of satisfaction to the reviewer. The followings are the authors' point-by-point responses:
Thank you very much.
Specific comments:
L20: reword cattle nutrition as cattle feeding
Response: The authors have rewritten cattle nutrition to cattle feeding (Line 20 and 36).
L 34: reword the deep study as an in-depth study
Response: The authors have replaced “deep study” with “in-depth study” (Line 35).
L40: remove extra space between “of” and “the”
Response: The authors have not seen the extra space between the “of” and “the” in the specified line and thus, the authors did not change anything.
L89: remove the phrase “that were”
Response: The authors have removed the phrase “that were”. Kindly refer to Line 90.
L102: give the appropriate reference for NDF and ADF methods used
Response: Thank you for the keen review. The authors have cited the references for the NDF and ADF methods used in this study. Kindly refer to Line 105.
L118: remove the term “all”
Response: The authors have removed the term “all”. Kindly refer to Line 121.
L152: remove the term “to”
Response: The authors have removed the term “to”. Kindly refer to Line 155.
L210-220 and in tables: Scientific names of Amaranth species should be in italic letters
Response: Thank you for the keen review. The authors have reformatted the scientific names of the amaranth plants. Kindly refer to Lines 222-224, 227-228, and 232-233.
Table 3: It would be better to mention VFA names (such as Acetate, propionate) instead of abbreviations used (C2, C3 etc)
Response: Thank you for the suggestion of the reviewer. As much as the authors would like to consider the reviewer’s comment, the authors opted to write the carbon numbers in the table for a neat and concise presentation of volatile fatty acids. Also, the authors have used the common names of the VFA in the footnote of the table and in the result and discussion section.
L277-278- give appropriate reference/s
Response: Line 277 was a statement made by the authors and hence, reference was not added. Instead, the line was re-written to “The lower NDF and ADF contents of the amaranths in this study signify higher non-fiber carbohydrates (NFC) such as CP, EE, ash, sugars, and starch” (Line 290-291) to clearly state that the findings was observed in the present study.
In line 228, the reference from where the statement was excerpted was already cited. However, the line was re-written for more clarity. Kindly refer to Lines 291-294.
L285, 287-288- The feed value of forage is a function of nutritive value and intake. Since the estimation of the feed value of the studied amaranth species was not an objective of the current experiment, the discussion on the “feed value” of Amaranth should be excluded. Please rephrase the sentence as appropriate.
Response: The authors appreciate the sound comment of the reviewer, and hence, the authors have re-written the term “feed value” to “feed quality”, which better aligned to the findings observed in this study. Kindly refer to following lines: 299 and 302. Also, a statement in the conclusion was added to state that the study only showed the potential value of amaranth as feed to cattle and further study was recommended to determine the true feed value of the amaranth on cattle (Lines 398-402).
L346: Referencing style should be uniform within the text
Response: Thank you for the keen review. The authors have reformatted the reference in the text. Kindly refer to Line 361.
Line363: Spelling mistake, please correct as “species”
Response: The authors have re-written the word specie to variety as “variety” is the more appropriate word for the samples used in this study. Kindly refer to Line 377-378.
Line370-373: Remove the discussion on antioxidant content and beef quality as those were not measured in the current study
Response: The authors have removed the discussion on antioxidant content and beef quality and have simplified the line to “The good to excellent feed quality of the amaranth on cattle could help in the utilization of uncommonly used forages” (Lines 389-390).
L381: remove the conclusions on feed value
Response: As much as the authors would like to consider the comment of the reviewer, the authors retained the statement in the conclusion section but has re-written the term “feed value” to “feed quality”, which gives a more appropriate inference of the study. Kindly refer to Line 398 for the revision made.
L448-453: please follow the correct referencing style
Response: Thank you for the keen review of the reviewer. The authors have corrected the reference style in the list (Lines 469-471) and in the text (Line 322).
Reviewer 3 Report
The authors have made a big effort during the review of the ms and some of my concerns have been addressed. I still feel that a better characterization, including a higher number of samples, should be made. However, considering the other reviewers’ comments, and the effort made by the authors in reviewing the ms., I could find the ms suitable for publication if a sentence is stated in the ms indicating that this paper presents first preliminary data and new studies should be conducted to properly evaluate the potential value of amaranth plants as feed forage for ruminant.
Author Response
Kindly see below point-by-point responses to the reviewer's comments or refer to the attached file. Thank you very much.
Reviewer No. 3 - Comments and Suggestions for Authors
General comments:
The authors have made a big effort during the review of the ms and some of my concerns have been addressed. I still feel that a better characterization, including a higher number of samples, should be made. However, considering the other reviewers’ comments, and the effort made by the authors in reviewing the ms., I could find the ms suitable for publication if a sentence is stated in the ms indicating that this paper presents first preliminary data and new studies should be conducted to properly evaluate the potential value of amaranth plants as feed forage for ruminant.
General response:
The authors are grateful for the appreciation of the reviewers to the efforts made in the previous revision of the manuscript. The authors have considered the suggestion of the reviewer and thus, statements were added in the conclusion section as follows:
This study presented the feed quality or the potential nutrient availability of the amaranth to cattle but did not present the feed value or the function of the nutritive value and intake of the amaranth to cattle. Hence, further study is recommended to entirely evaluate the value of the amaranth as feed forage for cattle.
The statements can be read in lines 398-402. Again, the authors are grateful for the thorough review of the reviewer. Thank you very much.
Round 2
Reviewer 2 Report
The authors have addressed my concerns appropriately in the revised version.
However, a minor text editing is recommended as follows.
The sentence in L381-383 seems redundant to be included in the conclusion. Rephrase the sentence in Line 383-384 as “However, further research is recommended to evaluate the feed value of Amaranth using cattle fed with Amaranth forage-based diets.”
Author Response
Please see the below response for the round 2 reviews of the reviewer or kindly see attached file for reference. Reviewer No. 2 (Round 2)
Comments and Suggestions for Authors
The authors have addressed my concerns appropriately in the revised version.
However, a minor text editing is recommended as follows.
The sentence in L381-383 seems redundant to be included in the conclusion. Rephrase the sentence in Line 383-384 as “However, further research is recommended to evaluate the feed value of Amaranth using cattle fed with Amaranth forage-based diets.”
Submission Date
25 November 2020
Date of this review
18 Dec 2020 09:04:49
Response:
Thank you for the follow-up review and the helpful review of the reviewer to make the conclusion section of the manuscript concise but informative. The authors have addressed the comment and rephrased Lines 441-445 to the suggestion of the reviewer. The sentence “However, further research is recommended to evaluate the feed value of Amaranth using cattle fed with Amaranth forage-based diets.” Can be found in Lines 445-446. Thank you very much.

This manuscript is a resubmission of an earlier submission. The following is a list of the peer review reports and author responses from that submission.
Round 1
Reviewer 1 Report
The work seems be properly conducted. Results are adequately described and discussed. However I do have a major concern.
The description of experimental unit is not totally clear and hence it is difficult to assess if the present study has sufficiently/enough data to perform an statistical analysis.
Sample collection do not declare how plant material was obtained
-was a single plant per especie-variety? if several plants were used, how many?
this is important because the experimental unit should be each plant sample as the author intentions should be to describe the potential nutritional values of the feed including the variation around this value.
currently it seems (based on the description) that a single sample obtained and the same sample was assayed several times. if this is the case, then the variation we obtain would be associated with the methodologies and the overall mean/average of all replicated provide an closer approximation to the true value of that particular sample, but by no mean provides could be associated to a plant population characterization which would allow compare especies and varieties as there would be no true replicate (experimental units) to begin with.
There other suggestion/comments included in the revised file

Reviewer 2 Report
This study provided some basic information for three different species of amaranth with their chemical composition and rumen fermentation characters by using in vitro and in situ methods. However, stage of harvest of the plants are missing. The most confusing thing is the CP content of the tested plants is quite low (lower than 5%, in table 1) while the authors highlighted as high CP content plant. Additionally, this plant can’t be comparable with corn silage which has high starch value.
Some minor comments:
Line 92-96 need give more detailed method from which AOAC methods (like code)
Line 104 more information for incubation system are required
Line 114 provide the information for pH detecting device
Line 125 Does the diet differ from that described in Line 101? Is 60% concentrate based on DM base?
Line 140 equation need to be corrected
Why not incubation for 72h for forage?
Line 335 to 338 Need to have the direct comparison in this study.
Reviewer 3 Report
The aim of the study was to evaluate the chemical composition, fermentation characteristics, and degradability of different species and subspecies of Amaranth. The study resulted very interesting and rich of data.
Some suggestions are reported below:
First of all, I suggest you review the guidelines of the journal, especially regarding the references within the text and at the end of the article.
Please check the English
Simple summary
Lines 15-18: Please change the period. The amaranth plants, one of the crops that can grow in poor soil and areas with extreme weather conditions – high temperature and limited rainfall, showed high potential feed value as forage for ruminants. An extensive study will help extend its utilization as alternative protein and fiber feed source in cattle nutrition.
Line 19: please delete namely. (A. caudatus, A. cruentus, and A. hypochondriacus).
Lines 20-23: please change the period. Rumen fluid was incubated with the amaranth plants in order to evaluate fermentation characteristics (total gas production, total volatile fatty acids concentration, pH, and ammonia concentration). There were no differences among the different amaranth species, but all amaranth showed favourable fermentation values.
Lines 23-26: please change the period. Suggestion: The effective degradability of dry matter and crude protein of the amaranth forages were also determined. Compared to other studies, the effective degradability of dry matter (33-56%) and crude protein (27-59%) of the amaranth were lower, these results could be due to the maturity stage at which the forages were harvested.
Lines 28-19: please simplify this period.
Line 31: please change suggest in suggesting.
Abstract
Lines 33-35: please change the sentence. The amaranth plants showed high potential feed value as forage for ruminants. A deep study of this plant, particularly in cattle, will help extend its utilization as alternative protein and fiber feed source in cattle nutrition.
Line 34: please delate namely
Line 37-40: please change the sentence. The in vitro technique was used in order to evaluate the fermentation characteristics such as total gas production, total volatile fatty acids (VFA) concentration, pH, and ammonia concentration of the rumen fluid. Moreover, the effective degradability of dry matter (EDDM), and crude protein (EDCP) of the amaranth forages were determined through in situ bag technique.
Introduction
The introduction is a bit short, perhaps you could expand the information on the nutritional characteristics and in particular of conservation. Here a reference: Jian Ma, Guoqing Sun, Ali Mujtaba Shah, Xue Fan, Shengli and Xiong Yu. Effects of Different Growth Stages of Amaranth Silage on the Rumen Degradation of Dairy Cows. Animals 2019, 9, 793
Lines 56-58: please change the sentences. The amaranth (Amaranthus spp.) is one of the crops that can grow in poor soil and areas with extreme weather conditions – high temperature and limited rainfall. These characteristics make it a valuable plant product particularly to parts of the world with shortage of water resources.
Line 58-61: please change the sentences. Moreover, the amaranth leaves and stems contained natural antioxidants such as vitamin C, phenolic acids, and flavonoids (1,2) which could increase colour stability in meat. Amaranth forages are excellent sources of fiber and protein which makes the amaranth a good alternative feed to livestock.
Line 60: please correct color in colour
Line 60: Please check the guidelines for the authors about the references. [1;2] not (1;2).
Line 64: reference number
Line 68: reference number
Line 66-67. Please change the sentence. Add some others references to the conservation methods.
Line 75: please change with to regarding
Material and methods
Lines 87-89: Please change the sentences. In this research three species and two subspecies for each sample were studied: A. caudatus (A.ca, 74 and 79), A. cruentus (A.cu 62 and 66), and A. hypochondriacus (A.hy 30 and 48).
Line 91: please correct thru
Line 94-95: please adjourn the refences about chemical composition to more recent years, in particular AOAC. Add the reference number
Line 98-103: please add reference
Line 104: add reference number
Line 105-110: please add reference
Line 118: please delete through the methods of
Line 125: please specific the ingredient of diet
Line 139: please add number reference
Line 152: maybe it’s better to use all replication to the statistical analysis (6x2x3=36)
Line 157: please insert the SEM, it is statistically more precise.
Results
Lines 164-165: please check this affirmation in relation to the data
Tab 1. Please define the percentage (% of what)
Lines 179-198: please rewrite the period, it isn’t clear, maybe try to give more space to table 2
Line 206: please change the sentences, it is no clear
Tab 3: please correct the editing. Moreover, maybe it’s better to insert the name of volatile fatty acids
Discussion
Please rewrite this whole part. Discussions written in this way are too confusing and combined with the results. it is necessary to justify the data obtained by comparing them with other articles and in perspective of what is stated in the purpose of the work.
Line 266: add reference number
Line 270: add reference number
Line 275: add reference number
Line 276: add reference number
Line 279: add reference number
Line 280: add reference number
Line 285: add reference number for all authors
Line 299: maybe, you could validate this sentence through a statistical correlation among the in vitro and in situ data
Line 303: add reference number
Reviewer 4 Report
This study aims to determine the nutritional value of the three different species of amaranth, namely A. caudatus, A. cruentus, and A. hypochondriacus, and the fermentation characteristics of the amaranth species through in vitro technology and in situ nylon bag study model. The subject of the paper is of interest since finding new forage sources is a priority in ruminants. However, on my point of view the data presented in the ms is insufficient to properly characterize amaranth spp, and consequently to determine its potential for livestock production. If I have understood correctly, only 3 samples of each subspecies have been analysed, which is clearly insufficient. Moreover, none information is provided about the plants (place, climate, age, stage of maturity, etc) or the difference between the 3 different runs (were all conducted with the same sample?). With the data provided here is it not possible to ensure that the characterization of amaranth species is fully correct and representative of the forage that will ge given to the animals.